# Identification of Novel Benzoxa-[2,1,3]-diazole Substituted Amino Acid Hydrazides as Potential Anti-Tubercular Agents

**DOI:** 10.3390/molecules24040811

**Published:** 2019-02-23

**Authors:** Alistair K. Brown, Ahmed K. B. Aljohani, Jason H. Gill, Patrick G. Steel, Jonathan D. Sellars

**Affiliations:** 1Institute for Cell and Molecular Biosciences, Faculty of Medical Sciences, Newcastle University, Catherine Cookson Building, Framlington Place, Newcastle upon Tyne NE2 4HH, UK; alistair.brown2@newcastle.ac.uk (A.K.B.); a.aljohani2@newcastle.ac.uk (A.K.B.A.); 2School of Pharmacy, Faculty of Medical Sciences, King George VI Building, Newcastle upon Tyne NE1 7RU, UK; 3Northern Institute for Cancer Research, Paul O′Gorman Building, Faculty of Medical Sciences, Newcastle University, Newcastle upon Tyne NE2 4HH, UK; jason.gill@newcastle.ac.uk; 4Department of Chemistry, Durham University, Lower Mountjoy, Stockton Road, Durham DH1 3LE, UK; p.g.steel@durham.ac.uk; 5Chemistry, School of Natural and Environmental Sciences, Newcastle University, Newcastle upon Tyne NE1 7RU, UK

**Keywords:** benzoxa-[2,1,3]-diazole, antibacterial, *Mycobacterium tuberculosis*, amino acid, hydrazide

## Abstract

Discovery and development of new therapeutic options for the treatment of *Mycobacterium tuberculosis* (*Mtb*) infection are desperately needed to tackle the continuing global burden of this disease and the efficacy and cost limitations associated with current medicines. Herein, we report the synthesis of a series of novel benzoxa-[2,1,3]-diazole substituted amino acid hydrazides in a two-step synthesis and evaluate their inhibitory activity against *Mtb* and selected bacterial strains of clinical importance utilising an end point-determined REMA assay. Alongside this, their potential for undesired cytotoxicity against mammalian cells was assessed employing standard MTT assay methodologies. It has been demonstrated using modification at three sites (the hydrazine, amino acid, and the benzodiazole) it is possible to change both the antibacterial activity and cytotoxicity of these molecules whilst not affecting their microbial selectivity, making them attractive architectures for further exploitation as novel antibacterial agents.

## 1. Introduction

Tuberculosis (TB) remains the predominant bacterial infectious disease globally, with approximately 10.4 million new cases and 1.7 million deaths recorded in 2016 [1]. Additionally, estimates suggest that over 20% of the population is latently infected with *Mycobacterium tuberculosis* (*Mtb*), the causative agent of TB [2]. Current treatment regimens of TB require patients to administer drug combinations for an extended period of time, at least six months, which is costly, prone to differential response rates, and exhibits significant problems with patient compliance and adherence [3]. The current treatment regimen involves combinations of rifampicin (RIF), isoniazid (INH), ethambutol (EMB), and pyrazinamide. However, resistance to these drug therapies has exacerbated the management issues of TB. In 2016 alone, almost 500,000 new cases of multidrug-resistant TB (resistant to both RIF and INH) and over 100,000 cases of RIF-resistant TB were identified [1]. Consequently, the identification of novel therapeutic options is an absolute necessity for the management of TB in the future [4]. In this context, in recent years several new drugs and/or regimens have been reported for TB, including bedaquiline [5], delamanid [6], clofazimine [7], bedazuiline-protomanid-linezolid [8]. Worryingly, resistance to several of these approaches has already been reported, primarily because of similar drug resistance mechanisms and target pathways [9,10]. Consequently, the development of new drug therapies for TB requires the discovery of new drug targets and novel structures which circumvent resistance mechanisms, whilst also enabling shorter treatment regimens.

Benzoxa-[2,1,3]-diazole, benzothia-[2,1,3]-diazole and benzotriazoles are a class of heterocycle which exhibit widespread therapeutic opportunities [11]. In addition, no studies have evaluated these compounds or their derivatives against *Mtb*. Inspired by virtual screening studies which identified simple substituted nitro benzoxa-[2,1,3]-diazoles as potential inhibitors of *Mtb* ATP phosphoribosyl transferase (HisG) and APS reductase (APSR) [12,13], we were attracted to this scaffold as a new starting point for the development of anti-TB drugs.

In this report we describe a series of novel benzoxa-[2,1,3]-diazole substituted amino acid hydrazides as selective drugs for the treatment of TB, highlighting the importance of the benzo-[2,1,3]-diazole, amino acid (AA) and the substituted aryl hydrazine (R_1_), towards *Mtb* selectivity, potency, efficacy, and avoidance of toxicity against mammalian cells (Figure 1).

## 2. Results and Discussion

In the context of a study to identify novel antibacterial agents designed to overcome antimicrobial resistance, a small library of diverse bioactive compounds had previously been synthesised within our team. Using the Resazurin Microtiter Assay (REMA) [14,15,16], these compounds were screened for antibacterial activity at a fixed concentration (128 μg/mL) against a range of drug-susceptible bacteria including Gram-positive, Gram-negative and *mycolata* bacteria (Appendix A) which revealed that many possessed little utility, even at these high concentrations. However, benzo-[2,1,3]-diazole architectures **1**–**12** were shown to possess antibacterial activity, including activity against *mycolata* bacteria and *Mtb* (Figure 2).

To gain an improved understanding of the antibacterial potency and scope of these compounds, a dose-range REMA assay was performed (128–0.125 μg/mL, converted to μM if active) (Table 1).

The results of the endpoint REMA assay revealed a mix of activity against the organisms tested, with simple substituted benzodiazole compounds **1**–**4** providing broad spectrum activity. Whilst the nitrobenzoxa-[2,1,3]-diazole **3** showed the highest levels of activity against *Mtb*, the lack of selectivity was a cause for concern. Replacing the nitro group with a sulphonamido amino acid moiety greatly improved specificity for *mycolata* bacteria, with substituted benzoxa-[2,1,3]-diazole **6** showing much higher activity than **5**, suggesting poor cell wall penetration of **5** is due to the carboxylic acid moiety. Notwithstanding this, replacement of the benzoxa-[2,1,3]-diazole with benzothia-[2,1,3]-diazoles **7** and **8** led to a complete loss of activity suggesting that the benzoxa-[2,1,3]-diazole plays a crucial role in these compounds antibacterial activity.

Further analysis of the results revealed that conversion of the ester to an aryl hydrazide **9**–**12** provided compounds more consistent activity across a range of structures. Consequently, substituted benzoxa-[2,1,3]-diazoles were chosen as the partner to amino acid hydrazides for further investigation, via a SAR study to further understand the importance of the amino acid (AA) and the hydrazine (R_1_) on anti-mycobacterial activity.

### 2.1. Chemical Synthesis of Benzoxa-[2,1,3]-diazole Amino Acid Hydrazides

To undertake this investigation, a two-step synthesis was engaged starting from *N*-Boc amino acids (Scheme 1). DCC coupling with a monosubstituted hydrazine produced the intermediate protected amino acid hydrazine which, following deprotection and condensation with benzoxa-[2,1,3]-diazole sulphonyl chloride afforded the desired products in moderate to good overall yields following flash chromatography purification (Table 2). Interestingly, several of the compounds exhibited restricted rotation as demonstrated by VT-^1^H-NMR spectroscopy (Appendix A).

### 2.2. Structure–Activity Relationships

Following the synthetic approach, Boc amino acid hydrazides **13a**–**22a** and benzoxa-[2,1,3]-diazole amino acid hydrazide **9**, **10**, **14b**–**22b** were screened in the same way against the same range of drug-susceptible bacteria as described above. Importantly, in line with the preliminary screening data these compounds showed high selectivity for *mycolata* bacteria (Appendix A). Focusing on the *Mtb* response, initially exploring the role of the amino acid, fixing the hydrazide and increasing the bulk of the amino acid substituent **13a**–**17a** resulted in diminished antibacterial activity of this component (Table 2). Subsequently, fixing the amino acid to glycine, we then evaluated the role of the hydrazine component (**18a**–**22a**). Introduction of an unsubstituted aromatic hydrazine **18a** alongside halogenated hydrazines **19a**–**22a** did not provide any significant enhancement in activity although a marked increase in cytotoxicity was observed. For both series, enhanced antibacterial activity was restored on coupling to the benzoxa-[2,1,3]-diazole **9**, **10**, **14b**–**22b** albeit at the cost of increased cytotoxicity, as noted for this subunit [17].

## 3. Discussion

Worryingly, as drug-resistant bacterial infections are on the rise and with the recent removal of antibiotic drug discovery programmes, there will be a significant demand for new chemical entities to address this condition. This study has identified that benzoxa-[2,1,3]-diazole substituted amino acid hydrazides have considerable potential as selective and potent agents against *Mtb*.

Throughout this study, the benzoxa-[2,1,3]-diazole core appears to be essential for activity. Whilst, as observed in some examples, the use of this unit is commonly associated with cytotoxicity this can be effectively modulated through the addition of the amino acid hydrazine. For example, we observed that conjugation with the amino acid hydrazides **19a** or **21a** provides a reduction in cytotoxicity (**19b**, **21b**, Table 2).

Excitingly, this modulation led to the use of a simple unsubstituted hydrazide **18a** which, although in isolation showed significant cytotoxicity, when conjugated with the benzoxadiazole **10** provides a compound with good level of activity against *Mtb* and no observable cytotoxicity.

## 4. Materials and Methods

### 4.1. Chemistry

#### 4.1.1. Synthesis of Hydrazides—General Procedure

A solution of *N*-Boc amino acid (0.25 g, 1.43 mmol, 1 equiv.), HOBt (2 equiv.) and DCC (1.2 equiv.) was dissolved in THF (7.5 mL), cooled to 0 °C and stirred for 15 min. The solution was treated with N-Aryl/Alkyl hydrazine * (1.2 equiv.) before warming to room temperature and stirring for a further 1.5 h. The mixture was then poured into sat. aq. NH_4_Cl (20 mL) before separating and extracting the aqueous layer with EtOAc (40 mL). The organic layer was further washed with sat. aq. NaHCO_3_ (20 mL) and then brine (20 mL). The combined organic layers were dried over MgSO_4_, filtered, concentrated and dried in vacuo. Flash chromatography (DCM/EtOH/NH_3_ [600:8:1], [400:8:1], [200:8:1]) afforded the desired *N*-Boc amino acid hydrazides.

* When the hydrazine hydrochlorides were used, Et_3_N (1.2 equiv.) is added to neutralise the salt

*tert*-Butyl 2-oxo-2-(2-(4-(trifluoromethyl)phenyl)hydrazinyl)ethylcarbamate, **13a**

Following the general procedure outlined, *N*-Boc-l-Glycine (0.50 g, 2.9 mmol) and 4-trifluorophenyl hydrazine (0.61 g, 3.5 MMOL) was transformed following flash chromatography into the title compound which was isolated as a white solid (0.45 g, 47%); m.p. 185–187 °C; υ_max_ (ATR) 3370 (NH), 3278 (NH), 3234 (NH), 3108, 3058, 2996, 1649 (C=O), 1615, 1518 (C=O), 1330, 1245, 1156, 1107, 1052, 828, 559 cm^−1^; δ_H_ (700 MHz, DMSO-d_6_) 9.78 (1H, s, Ar-N*H*NH), 8.32 (1H, s, Ar-NHN*H*), 7.40 (2H, d, *J* 9, Ar-*H*), 7.07 (1H, t, *J* 6, BocN*H*), 6.78 (2H, d, *J* 9, Ar-*H*), 3.60 (2H, d, *J* 6, NHC*H*_2_), 1.37 (9H, s, (C*H*_3_)_3_COC(O)NH); δ_C_ (176 MHz, DMSO-d_6_) 170.2 (*C*ONH), 156.6 ((CH_3_)_3_CO*C*(O)NH), 153.1 (Ar-*C*), 126.7 (Ar-*C*), 126.4 (Ar-*C*), 118.82 (Ar-*C*), 118.63 (Ar-*C*), 112.1 (Ar-*C*), 78.8 (NH*C*H_2_), 42.7 ((CH_3_)_3_*C*O), 28.8 ((*C*H_3_)_3_CO); δ_F_ (658 MHz, DMSO-d_6_) -59.30; *m*/*z* (ES^+^) 356 (MNa^+^), 689 (2M + Na^+^); HRMS (ES^+^) Found MH^+^, 334.13722 (C_14_H_19_F_3_N_3_O_3_ requires 334.13730).

*tert*-Butyl (*S*)-(2-oxo-1-phenyl-2-(2-(4-(trifluoromethyl)phenyl)hydrazineyl)ethyl)carbamate, **14a**

Following the general procedure outlined, *N*-Boc-l-phenylglycine (0.250 g, 0.99 mmol) and 4-trifluorophenyl hydrazine (0.30 g, 1.75 mmol) was transformed following flash chromatography into the title compound as a brown solid (0.386 g, 95%); R_f_ 0.44 (DCM/EtOH/NH_3_ [200:8:1]); m.p. 138–143 °C; υ_max_ (ATR) 3328 (N-H), 2982, 2930, 2851, 1678 (C=O), 1655 (C=O), 1524, 1322, 1158, 1113, 1067, 834, 696, 558, 488 cm^−1^; δ_H_ (400 MHz, CDCl_3_) 8.00 (1H, bs, N*H*), 7.46–7.37 (7H, m, Ar-*H*), 6.68 (2H, d, *J* 8, Ar-*H*), 6.24 (1H, bs, N*H*), 5.69 (1H, d, *J* 7, NHC*H*CO), 5.35 (1H, bs, N*H*), 1.42 (9H, s, C(C*H*_3_)_3_); δ_C_ (176 MHz, CDCl_3_) 157.0 (NHCH*C*O), 150.3 (NH*C*OO), 129.3 (Ar-*C*), 128.9 (Ar-*C*), 128.4 (Ar-*C*), 127.2 (Ar-*C*), 127.1 (Ar-*C*), 126.5 (Ar-*C*), 125.1 (Ar-*C*), 123.6 (*C*F_3_), 112.7 (Ar-*C*), 80.3 (*C*(CH_3_)_3_), 59.4 (NH*C*HCO), 28.3 (C(*C*H_3_)_3_); δ_F_ (376 MHz, CDCl_3_) -61.60 (3F, s, C*F*_3_); *m*/*z* (ES^+^) 410 (MH^+^), 432 (MNa^+^), 841 (2M + Na^+^); HRMS (ES^+^) Found MH^+^, 410.1689 (C_20_H_23_F_3_N_3_O_3_ requires 410.1686).

*tert*-Butyl (*S*)-2-(2-(4-(trifluoromethyl)phenyl)hydrazine-1-carbonyl)pyrrolidine-1-carboxylate, **15a**

Following the general procedure outlined, *N*-Boc-l-proline (0.250 g, 1.16 mmol) and 4-trifluorophenyl hydrazine (0.30 g, 1.75 mmol) was transformed following flash chromatography into the title compound as a golden-brown solid (0.366 g, 85%); R_f_ 0.38 (DCM/EtOH/NH_3_ [200:8:1]); m.p. 116–120 °C; υ_max_ (ATR) 3265 (N-H), 2979, 2932, 2851, 1672 (C=O), 1618, 1399, 1324, 1107, 1065, 833, 591 cm^−1^; δ_H_ (400 MHz, CDCl_3_) 8.91 (1H, bs, N*H*), 7.45 (2H, d, *J* 8, Ar-*H*), 6.87 (2H, d, *J* 8, Ar-*H*), 6.34 (1H, bs, N*H*), 4.45–4.34 (1H, m, NC*H*CO), 3.52–3.35 (2H, m, N(C*H*_2_)_3_), 2.06–1.87 (4H, m, N(C*H*_2_)_3_), 1.53 (9H, s, C(C*H*_3_)_3_); δ_C_ (176 MHz, CDCl_3_) 172.2 (NCH*C*O), 156.1 (N*C*OO), 150.8 (Ar-*C*), 126.4 (Ar-*C*), 125.2 (Ar-*C*), 123.7 (*C*F_3_), 112.7 (Ar-*C*), 81.0 (*C*(CH_3_)_3_), 58.4 (N*C*HCO), 47.2 (N*C*H_2_(CH_2_)_2_), 28.4 (C(*C*H_3_)_3_), 25.5, (NCH_2_(*C*H_2_)_2_), 24.8 (NCH_2_(*C*H_2_)_2_); *m*/*z* (ES^+^) 374 (MH^+^), 396 (MNa^+^), 769 (2M + Na^+^); HRMS (ES^+^) Found MNa^+^, 396.1497 (C_17_H_22_F_3_N_3_O_3_Na requires 396.1505).

*tert*-Butyl (*S*)-(1-oxo-3-phenyl-1-(2-(4-(trifluoromethyl)phenyl)hydrazineyl)propan-2-yl)carbamate, **16a**

Following the general procedure outlined, *N*-Boc-l-phenylalanine (0.250 g, 0.94 mmol) and 4-trifluorophenyl hydrazine (0.30 g, 1.75 mmol) was transformed following flash chromatography into the title compound as a golden-brown solid (0.330 g, 83%); R_f_ 0.39 (DCM/EtOH/NH_3_ [200:8:1]); m.p. 157–159 °C; υ_max_ (ATR) 3324 (N-H), 3274 (N-H), 2931, 2851, 1686 (C=O), 1660 (C=O), 1520, 1328, 1156, 1103, 1067, 835, 716, 515 cm^−1^; δ_H_ (400 MHz, CDCl_3_) 7.93 (1H, bs, N*H*), 7.39 (2H, d, *J* 8, Ar-*H*), 7.36–7.32 (3H, m, Ar-*H*), 7.26–7.22 (2H, m, Ar-*H*), 6.61 (2H, d, *J* 8, Ar-*H*), 6.18 (1H, bs, N*H*), 5.09 (1H, bd, *J* 7, N*H*), 4.46 (1H, q, *J* 8, CH_2_C*H*CO), 3.12 (2H, d, *J* 8, C*H*_2_CHCO), 1.47 (9H, s, C(C*H*_3_)_3_); δ_C_ (176 MHz, CDCl_3_) 171.6 (NHCH*C*O), 150.3 (NH*C*OO), 135.96 (Ar-*C*), 129.3 (Ar-*C*), 128.9 (Ar-*C*), 127.2 (Ar-*C*), 126.4 (Ar-*C*), 126.4 (Ar-*C*), 125.2 (Ar-*C*), 123.6 (*C*F_3_), 112.7 (Ar-*C*), 80.9 (*C*(CH_3_)_3_), 54.5 (NH*C*HCO), 33.5 (*C*H_2_CHCO), 28.3 (C(*C*H_3_)_3_); δ_F_ (376 MHz, CDCl_3_) -61.56 (3F, s, C*F*_3_); *m*/*z* (ES^+^) 424 (MH^+^), 446 (MNa^+^), 869 (2M + Na^+^); HRMS (ES^+^) Found MH^+^, 424.1847 (C_21_H_25_F_3_N_3_O_3_ requires 424.1843).

*tert*-Butyl (*S*)-(1-oxo-1-(2-(4-(trifluoromethyl)phenyl)hydrazineyl)propan-2-yl)carbamate, **17a**

Following the general procedure outlined, *N*-Boc-l-alanine (0.250 g, 1.32 mmol) and 4-trifluorophenyl hydrazine (0.30 g, 1.75 mmol) was transformed following flash chromatography into the title compound as a dark brown solid (0.425 g, 93%); R_f_ 0.37 (DCM/EtOH/NH_3_ [200:8:1]); m.p. 126–131 °C; υ_max_ (ATR) 3321 (N-H), 2929, 2851, 1683 (C=O), 1664 (C=O), 1617, 1524, 1322, 1157, 1101, 1066, 830, 641 cm^−1^; δ_H_ (400 MHz, CDCl_3_) 8.50 (1H, bs, N*H*), 7.46 (2H, d, *J* 8, Ar-*H*), 6.87 (2H, d, *J* 8, Ar-*H*), 6.34 (1H, bs, N*H*), 5.07 (1H, bd, *J* 6, N*H*), 4.40 (1H, m, CH_3_C*H*), 1.50 (9H, s, C(C*H*_3_)_3_), 1.43 (3H, d, *J* 7, C*H*_3_CH); δ_C_ (176 MHz, CDCl_3_) 171.7 (NHCH*C*O), 158.7 (NH*C*OO), 129.3 (Ar-*C*), 128.9 (Ar-*C*), 127.2 (Ar-*C*), 126.4 (*C*F_3_), 112.8 (Ar-*C*), 80.9 (*C*(CH_3_)_3_), 54.6 (NH*C*HCO), 33.7 (CH*C*H_3_), 28.3 (C(*C*H_3_)_3_); δ_F_ (376 MHz, CDCl_3_) -61.55 (3F, s, C*F*_3_); *m*/*z* (ES^+^) 348 (MH^+^), 370 (MNa^+^), 717 (2M + Na^+^); HRMS (ES^+^) Found MNa^+^, 370.1352 (C_15_H_20_F_3_N_3_O_3_Na requires 370.1349).

*tert*-Butyl 2-oxo-2-(2-phenylhydrazinyl)ethylcarbamate, **18a**

Following the general procedure outlined, *N*-Boc-l-Glycine (0.50 g, 2.9 mmol) and phenyl hydrazine (0.30 mL, 3.1 mmol) was transformed following flash chromatography into the title compound as a white solid (0.30 g, 39%); R_f_ 0.1 (n-hexane/EtOAc 7:3) as a mixture of rotamers in the ratio [5:1] by NMR @ 25 °C; m.p. 120–122 °C; υ_max_ (ATR) 3348 (NH), 3274 (NH), 2922, 2852, 1746, 1652 (C=O), 1640 (C=O), 1519, 1488, 1418, 1375, 1260, 1192, 1106, 1036, 874, 760 cm^−1^; δ_H_ (700 MHz, DMSO-d_6_) 9.59 (1H, s, N*H*NHPh), 9.08 (0.2H, s, N*H*NHPh), 7.85 (0.22H, s, NHN*H*Ph), 7.65 (1H, s, NHN*H*Ph), 7.17 (0.45H, t, *J* 8, Ar-*H*), 7.08 (2H, t, *J* 8, Ar-*H*), 7.02 (1H, t, *J* 5, BocN*H*), 6.74 (0.23H, t, *J* 8, Ar-*H*), 6.71 –6.63 (3H, m, Ar-H), 6.61 (0.23H, m, Ar-H), 3.68 (0.34H, d, *J* 6, BocNHC*H*_2_), 3.58 (2H, d, *J* 5, BocNHC*H*_2_), 1.37 (9H, s, (C*H*_3_)_3_COC(O)NH), 1.34 (1.33H, s, (C*H*_3_)_3_COC(O)NH); δ_C_ (176 MHz, DMSO-d_6_) 174.2 (*C*ONH), 169.9 (*C*ONH), 156.5 ((CH_3_)_3_CO*C*(O)NH), 149.9 (Ar-*C*), 149.1 (Ar-*C*), 129.3 (Ar-*C*), 119.0 (Ar-*C*), 112.8 (Ar-*C*), 112.6 (Ar-*C*), 78.7 (NH*C*H_2_), 78.5 (NH*C*H_2_), 42.7 ((CH_3_)_3_*C*O), 31.9 ((CH_3_)_3_*C*O), 28.8 ((*C*H_3_)_3_CO), 22.8 ((*C*H_3_)_3_CO); *m*/*z* (ES^+^) 266 (MH^+^), 531 (2M + H^+^); HRMS (ES^+^) Found MH^+^, 266.24985 (C_13_H_20_O_3_N_3_ requires 266.14992).

*tert*-Butyl (2-(2-(4-fluorophenyl)hydrazineyl)-2-oxoethyl)carbamate, **19a**

Following the general procedure outlined, *N*-Boc-l-Glycine (0.25 g, 1.45 mmol) and 4-fluorophenyl hydrazine (0.20 g, 1.57 mmol) was transformed following flash chromatography (DCM/EtOH/NH_3_ [600:8:1], [400:8:1], [200:8:1]) into the title compound as a light brown solid (0.32 g, 78%) as a mixture of rotamers [4:1] by NMR @ 25 °C; R_f_ 0.32 (DCM/EtOH/NH_3_ [200:8:1]); m.p. 127–129 °C; υ_max_ (ATR) 3364 (NH), 3269 (NH), 3132, 2984, 1652 (C=O), 1505, 1393, 1370, 1269, 1225, 1224, 1162, 1029, 1007 cm^−1^; NMR data given for major rotamer δ_H_ (700 MHz, CDCl_3_) 8.35 (1H, bs, (CH_3_)_3_COC=ON*H*), 6.89 (2H, t, *J* 17, Ar-*H*), 6.75 (2H, dd, *J* 9, 4, Ar-*H*), 5.30 δ1H, bs, N*HC*=O), 4.07 (1H, bs, N*H*), 3.85 (2H, d, *J* 6, C*H*_2_NH), 1.45 (9H, s, (C*H*_3_)_3_COC=ONH); δ_C_ (176 MHz, CDCl_3_) 170.4 (*C*=O), 157.4 (*C*=O), 158.6 (Ar-*C*) 143.7 (Ar-*C*NH), 115.7 (Ar-*C*), 114.9 (Ar-*C*), 81.0 ((CH_3_)_3_*C*OC=ONH) 43.5 (*C*H_2_), 28.7 ((*CH*_3_)_3_COC=ONH); δ_F_ (376MHz, CDCl_3_) -123.19 (Ar-*F*); *m*/*z* (ES^+^) 284 (MH^+^), 306 (MNa^+^), 589 (2M + Na^+^); HRMS (ES^+^) Found MH^+^, 284.1412 (C_13_H_19_N_3_O_3_F requires 284.1410).

*tert*-Butyl (2-(2-(4-chlorophenyl)hydrazineyl)-2-oxoethyl)carbamate, **20a**

Following the general procedure outlined, *N*-Boc-l-Glycine (0.25 g, 1.45 mmol) and 4-chlorophenyl hydrazine (0.22 g, 1.57 mmol) was transformed following flash chromatography into the title compound as a brown solid (0.41 g, 96%) as a mixture of rotamers [4:1] by NMR @ 25 °C; R_f_ 0.2 (*n*-hexane/EtOAc [1:1]); m.p. 118–120 °C; υ_max_ (ATR) 3365, 3275 (NH), 3122, 3044, 2929, 2851, 1697, 1650 (C=O), 1525, 1491, 1441, 1391, 1369, 1311, 1268, 1241, 1158, 1091, 1053 cm^−1^; NMR data given for major rotamer δ_H_ (700 MHz, CDCl_3_) 8.07 (1H, bs, C=ON*H*), 7.17 (2H, d, *J* 9, Ar-*H*), 6.76 (2H, d, *J* 9, Ar-*H*), 5.16 (1H, bs, N*H*), 3.87 (2H, d, *J* 7, C*H*_2_NH), 3.45 (1H, bs, N*H*), 1.47 (9H, s, (C*H*_3_)_3_COC=ONH); δ_C_ (176 MHz, CDCl_3_) 169.0 (*C*=O), 156.5 (*C*=O), 146.2 (Ar-*C*Cl), 129.1 (Ar-*C*), 126.2 (Ar-*C*NH), 114.8 (Ar-*C*), 80.4 ((CH_3_)_3_*C*OC=ONH), 43.8 (*C*H_2_NH), 28.3 ((*C*H_3_)_3_COC=ONH); *m*/*z* (ES^+^) 300 ([^35^Cl]MH^+^), 302 ([^37^Cl]MH^+^), 322 ([^35^Cl]MNa^+^), 324 ([^37^Cl]MNa^+^), 623 ([^35,35^Cl]2M+Na^+^), 625 ([^35,37^Cl]2M + Na^+^), 627 ([^37,37^Cl]2M + Na^+^); HRMS (ES^+^) Found [^35^Cl]MH^+^, 300.1134 (C_13_H_19_N_3_O_3_^35^Cl requires 300.1115).

*tert*-Butyl (2-(2-(3-chlorophenyl)hydrazineyl)-2-oxoethyl)carbamate, **21a**

Following the general procedure outlined, *N*-Boc-l-Glycine (0.25 g, 1.45 mmol) and 3-chlorophenyl hydrazine hydrochloride (0.28 g, 1.57 mmol) was transformed following flash chromatography into the title compound as a pale brown solid (0.34 g, 80%) as a mixture of rotamers [4:1] by NMR @ 25 °C; R_f_ 0.38 (DCM/EtOH/NH_3_ [200:8:1]); m.p. 119.5–121.7 °C; υ_max_ (ATR) 3343 (NH), 3269 (NH), 3077, 2980, 2932, 1659 (C=O), 1597, 1512, 1490, 1368, 1242, 1230, 1156, 1046, 1030 cm^−1^; NMR data given for major rotamers δ_H_ (700 MHz, CDCl_3_) 8.09 (1H, bs, N*H*), 7.12 (1H, t, *J* 8, Ar-*H*), 6.85 (1H, ddd, *J* 8, 2, 1, Ar-*H*), 6.81 (1H, dd, *J* 2, Ar-*H*), 6.70 (1H, ddd, *J* 8, 2, 1, Ar-*H*), 5.17 (1H, bs, N*H*), 3.88 (2H, d, *J* 6, C*H*_2_NH), 1.47 (9H, s, (C*H*_3_)_3_COC=ONH); δ_C_ (176 MHz, CDCl_3_) 170.0 (*C*=O), 157.5 (*C*=O), 148.9(Ar-*C*), 135.1 (Ar-*C*NH), 130.2 (Ar-*C*), 121.3 (Ar-*C*), 113.4 (Ar-*C*) 111.8 (Ar-*C*), 81.2 ((CH_3_)_3_*C*OC=ONH), 28.3 ((*C*H_3_)_3_COC=ONH); *m*/*z* (ES^+^) 300 ([^35^Cl]MH^+^), 302 ([^37^Cl]MH^+^), 322 ([^35^Cl]MNa^+^), 324 ([^37^Cl]MNa^+^), 623 ([^35,35^Cl]2M + Na^+^), 625 ([^35,37^Cl]2M + Na^+^), 627 ([^37,37^Cl]2M + Na^+^); HRMS (ES^+^) Found [^35^Cl]MH^+^, 300.1119 (C_13_H_19_N_3_O_3_^35^Cl requires 300.1115).

*tert*-Butyl (2-(2-(2-chlorophenyl)hydrazineyl)-2-oxoethyl)carbamate, **22a**

Following the general procedure outlined, *N*-Boc-l-Glycine (0.25 g, 1.45 mmol) and 2-chlorophenyl hydrazine hydrochloride (0.28 g, 1.57 mmol) was transformed following flash chromatography into the title compound as a light brown gum (0.41 g, 95%) as a mixture of rotamers [3:1] by NMR @ 25 °C; R_f_ 0.50 (DCM/EtOH/NH_3_ [200:8:1]); υ_max_ (ATR) 3262 (NH), 3074, 2960, 2928, 2851, 2118, 1668 (C=O), 1593, 1536, 1498, 1456, 1393, 1365, 1281, 1254, 1226, 1171, 1053, 1035 cm^−1^; NMR data given for major rotamer δ_H_ (700 MHz, CDCl_3_) 8.23 (1H, bs, N*H*), 7.31 (1H, d, *J* 8, Ar-*H*), 7.14 (1H, t, *J* 8, Ar-*H*), 6.88 (1H, dd, *J* 8, *J* 1, Ar-*H*), 6.84 (1H, t, *J* 8, Ar-*H*), 6.40 (1H, bs, N*H*), 3.90 (2H, d, *J* 6, C*H*_2_NH), 3.47 (1H, bs, N*H*), 1.47 (9H, s, (C*H*_3_)_3_COC=ONH); δ_C_ (176 MHz, CDCl_3_) 169.7 (*C*=O), 156.9 (*C*=O), 143.5 (Ar-*C*Cl), 129.5 (Ar-*C*), 127.6 (Ar-*C*), 121.4 (Ar-*C*), 119.5 (Ar-*C*NH), 113.5 (Ar-*C*), 80.9 ((CH_3_)_3_*C*OC=ONH), 43.5 (*C*H_2_NH), 28.2 ((*C*H_3_)_3_COC=ONH); *m*/*z* (ES^+^) 300 ([^35^Cl]MH^+^), 302 ([^37^Cl]MH^+^), 322 ([^35^Cl]MNa^+^), 324 ([^37^Cl]MNa^+^), 623 ([^35,35^Cl]2M + Na^+^), 625 ([^35,37^Cl]2M + Na^+^), 627 ([^37,37^Cl]2M + Na^+^); HRMS (ES^+^) Found [^35^Cl]MH^+^, 300.1108 (C_13_H_19_N_3_O_3_^35^Cl requires 300.1115).

#### 4.1.2. Synthesis of Benzoxa-[2,1,3]-Diazole Peptidomimetics—General Procedure

*N*-Boc amino acid hydrazides (1.20 equiv.) were dissolved in 4 M HCl solution in dioxane (3 mL) and stirred for 30 min at room temperature. The solvent was then removed in vacuo*,* the resulting solid was suspended in THF (3 mL) and Et_3_N (3 equiv.) was added. The solution was cooled to 0 °C, before treating with 7-chlorobenzoxa-[2,1,3]-diazole-4-sulphonyl chloride (50 mg, 0.20 mmols, 1 equiv.) before warming to room temperature and stirring for 2 h. The mixture was then poured into sat. aq. NH_4_Cl (20 mL). The aqueous layer was separated and extracted with EtOAc (20 mL). The combined organic layers were dried over MgSO_4_, filtered, concentrated and dried in vacuo. Flash chromatography (DCM/EtOH/NH_3_ [600:8:1], [400:8:1], [200:8:1]) afforded the desired sulphonamides.

7-Chloro-N-(2-oxo-2-(2-(4-(trifluoromethyl)phenyl)hydrazinyl)ethyl)benzo[c][1,2,5]oxadiazole-4-sulfonamide, **9**

Following the general procedure outlined, *tert*-Butyl 2-oxo-2-(2-(4-(trifluoromethyl)phenyl)hydrazinyl)ethylcarbamate, **13a** (79 mg, 0.24 mmol) was transformed following flash chromatography into the title compound as a pale brown solid (6 mg, 7%); R_f_ 0.6 (DCM/MeOH 9:1) as a mixture of rotamers in the ratio [7:1] by NMR @ 25 °C; m.p. 211–213 °C; υ_max_ (ATR) 332, 3158, 1685 (C=O), 1614, 1524, 1473, 1418, 1326, 1155, 1107, 1062, 952, 834, 632 cm^−1^; δ_H_ (500 MHz, DMSO-d_6_) 9.88 (1H, s, Ar-N*H*NH), 9.31 (0.13H, s, Ar-N*H*NH), 8.73 (1H, t, *J* 6, SO_2_N*H*), 8.46 (0.14H, s, Ar-NHN*H*), 8.41 (0.14H, t, *J* 6, SO_2_N*H*), 8.26 (1H, s, Ar-NHN*H*), 8.02 (1H, d, *J* 8, Ar-*H*), 7.95 (0.16H, d, *J* 7, Ar-*H*), 7.91 (0.17H, d, *J* 7, Ar-*H*), 7.85 (1H, d, *J* 8, Ar-*H*), 7.57 (0.34H, d, *J* 8, Ar-*H*), 7.43 (2H, d, *J* 8, Ar-*H*), 6.79 (0.32H, d, *J* 8, Ar-*H*), 6.68 (2H, d, *J* 8, Ar-*H*), 3.94 (0.25H, d, *J* 6, NHC*H*_2_), 3.88 (2H, d, *J* 6, NHC*H*_2_); δ_C_ (126 MHz, DMSO-d_6_) 168.4 (*C*=O), 152.6 (Ar-*C*), 149.4 (Ar-*C*), 146.2 (Ar-*C*), 134.5 (Ar-*C*), 131.5 (Ar-*C*), 129.1 (Ar-*C*), 126.7 (Ar-*C*), 125.6 (Ar-*C*), 112.1 (Ar-*C*), 44.3 (NH*C*H_2_); δ_F_ (376 MHz, DMSO) -59.71, -59.89; *m*/*z* (ES^−^) 448 ([^35^Cl]M^-^), 450 ([^37^Cl]M^−^), 897 ([^35^Cl,^35^Cl]2M^−^), 899 ([^35^Cl,^37^Cl]2M^−^), 900 ([^37^Cl,^37^Cl]2M^−^); HRMS (ES^−^) Found [^35^Cl]M^−^, 448.00931 (C_15_H_10_^35^ClF_3_N_5_O_4_S requires 448.00996).

(*S*)-7-Chloro-*N*-(2-oxo-1-phenyl-2-(2-(4-(trifluoromethyl)phenyl)hydrazineyl)ethyl)benzo[c][1,2,5]oxadiazole-4-sulfonamide, **14b**

Following the general procedure outlined, *tert*-butyl (*S*)-(2-oxo-1-phenyl-2-(2-(4-(trifluoromethyl)phenyl)hydrazineyl)ethyl)carbamate, **14a** (97 mg, 0.24 mmol) was transformed following flash chromatography into the title compound as a pale yellow solid (66 mg, 63%); R_f_ 0.46 (DCM/EtOH/NH_3_ [200:8:1]); m.p. 183–187 °C; υ_max_ (ATR) 3275 (N-H), 2930, 1693 (C=O), 1618, 1521, 1325, 1161, 1105, 1067, 840, 800, 636, 619, 590, 570, 530, 504, 491 cm^−1^; δ_H_ (700 MHz, DMSO-d_6_) 10.20 (1H, s, N*H*), 9.35 (1H, bs, N*H*), 8.36 (1H, s, N*H*), 7.97 (1H, d, *J* 7, Ar-*H*), 7.82 (1H, d, *J* 7, Ar-*H*), 7.32–7.27 (4H, m, Ar-*H*), 7.18–7.15 (3H, m, Ar-*H*), 6.44 (2H, d, *J* 8, Ar-*H*), 5.30 (1H, s, NHC*H*CO); δ_C_ (176 MHz, DMSO-d_6_) 168.9 (NHCH*C*O), 152.3 (Ar-*C*), 148.9 (Ar-*C*NO), 145.3 (Ar-*C*NO), 137.1 (Ar-*C*), 134.9 (Ar-*C*), 131.1 (Ar-*C*), 128.9 (Ar-*C*), 128.5 (Ar-*C*), 128.3 (Ar-*C*), 127.9 (Ar-*C*),127.7 (Ar-*C*), 126.4 (Ar-*C*), 125.6 (Ar-*C*), 124.5 (*C*F_3_), 111.6 (Ar-*C*), 59.0 (NH*C*HCO); *m*/*z* (ES^+^) 526 ([^35^Cl] MH^+^), 528 ([^37^Cl] MH^+^), 548 ([^35^Cl] MNa+), 550 ([^37^Cl] MNa^+^), 1073 ([^35,35^Cl] 2M + Na^+^), 1075 ([^35,37^Cl] 2M + Na^+^), 1077 ([^37,37^Cl] 2M+Na^+^); HRMS (ES^+^) Found [^35^Cl]MNa^+^, 526.0565 (C_21_H_16_F_3_N_5_O_4_S^35^Cl requires 526.0558).

(*S*)-1-((7-Chlorobenzo[c][1,2,5]oxadiazol-4-yl)sulfonyl)-N′-(4-(trifluoromethyl)phenyl)pyrrolidine-2-carbohydrazide, **15b**

Following the general procedure outlined, *tert*-butyl (*S*)-2-(2-(4-(trifluoromethyl)phenyl)hydrazine-1-carbonyl)pyrrolidine-1-carboxylate, **15a** (89 mg, 0.24 mmol) was transformed following flash chromatography into the title compound as a yellow solid (82 mg, 85%); R_f_ 0.46 (DCM/EtOH/NH_3_ [200:8:1]); m.p. 166–170 °C; υ_max_ (ATR) 3324 (N-H), 2929, 2851, 1682 (C=O), 1621, 1574, 1325, 1146, 1102, 1066, 946, 836, 616 cm^−1^; δ_H_ (700 MHz, DMSO-d_6_) 10.09 (1H, s, N*H*), 8.39 (1H, bs, N*H*), 8.08 (1H, d, *J* 7, Ar-*H*), 7.86 (1H, d, *J* 7, Ar-*H*), 7.44 (2H, d, *J* 9, Ar-*H*), 6.82 (2H, d, *J* 9, Ar-*H*), 4.43 (1H, dd, *J* 9, 4, NC*H*CO), 3.61–3.57 (1H, ddd, *J* 10, 7, 5, NC*H*_2_(CH_2_)_2_), 3.47–3.41 (1H, dt, *J* 10, 7, NC*H*_2_(CH_2_)_2_), 1.26–0.96 (4H, m, NCH_2_(C*H*_2_)_2_); δ_C_ (176 MHz, DMSO-d_6_) 171.4 (NHCH*C*O), 152.8 (Ar-*C*), 149.5 (Ar-*C*NO), 146.3 (Ar-*C*NO), 136.8 (Ar-*C*), 131.2 (Ar-*C*), 127.7 (Ar-*C*), 126.5 (Ar-*C*), 126.3 (Ar-*C*), 125.5 (Ar-*C*), 124.7 (*C*F_3_), 111.9 (Ar-*C*), 60.7 (N*C*HCO), 49.3 (N*C*H_2_(CH_2_)_2_), 33.9 (NCH_2_(*C*H_2_)_2_), 24.9 (NCH_2_(*C*H_2_)_2_); *m*/*z* (ES^+^) 490 ([^35^Cl] MH^+^), 492 ([^37^Cl] MH^+^); HRMS (ES^+^) Found [^35^Cl]MH^+^, 490.0555 (C_18_H_16_F_3_N_5_O_4_S^35^Cl requires 490.0558).

(*S*)-7-Chloro-*N*-(1-oxo-3-phenyl-1-(2-(4-(trifluoromethyl)phenyl)hydrazineyl)propan-2-yl)benzo[c][1,2,5]oxadiazole-4-sulfonamide, **16b**

Following the general procedure outlined *tert*-butyl (*S*)-(1-oxo-3-phenyl-1-(2-(4-(trifluoromethyl)phenyl)hydrazineyl)propan-2-yl)carbamate, **16a** (100 mg, 0.24 mmol) was transformed following flash chromatography into the title compound which was isolated as a brown solid (91 mg, 85%); R_f_ 0.41 (DCM/EtOH/NH_3_ [200:8:1]); m.p. 80–84 °C; υ_max_ (ATR) 3321 (N-H), 2932, 2854, 1677 (C=O), 1617, 1323, 1156, 1109, 1065, 834, 700, 632, 618, 577, 488 cm^−1^; δ_H_ (700 MHz, DMSO-d_6_) 10.13 (1H, s, N*H*), 9.06 (1H, d, *J* 9, N*H*), 8.43 (1H, bs, N*H*), 7.76 (1H, d, *J* 7, Ar-*H*), 7.71 (1H, d, *J* 7, Ar-*H*), 7.41 (2H, d, *J* 9, Ar-*H*), 7.00–6.96 (2H, m, Ar-*H*), 6.84–6.79 (3H, m, Ar-*H*), 6.70 (2H, d, *J* 9, Ar-*H*), 4.16–4.12 (1H, m, CH_2_C*H*CO), 2.87 (1H, dd, *J* 14, 4, C*H*_2_CHCO), 2.68 (1H, dd, *J* 14, 11, C*H*_2_CHCO); δ_C_ (176 MHz, DMSO-d_6_) 170.9 (NHCH*C*O), 152.6 (Ar-*C*), 148.9 (Ar-*C*NO), 144.7 (Ar-*C*NO), 136.9 (Ar-*C*), 133.9 (Ar-*C*), 130.7 (Ar-*C*), 129.5 (Ar-*C*), 128.2 (Ar-*C*), 127.6 (Ar-*C*), 126.5 (Ar-*C*), 126.2 (Ar-*C*), 126.2 (Ar-*C*), 125.5 (Ar-*C*), 124.6 (*C*F_3_), 111.8 (Ar-*C*), 57.4 (NH*C*HCO), 38.0 (*C*H_2_CHCO); *m*/*z* (ES^+^) 540 ([^35^Cl] MH^+^), 542 ([^37^Cl] MH^+^), 562 ([^35^Cl] MNa^+^), 564 ([^37^Cl] MNa^+^), 1101 ([^35,35^Cl] 2M + Na^+^), 1103 ([^35,37^Cl] 2M + Na^+^), 1105 ([^37,37^Cl] 2M + Na^+^). HRMS (ES^+^) Found [^35^Cl]MH^+^, 540.0642 (C_22_H_18_^35^ClF_3_N_5_O_4_S requires 540.0715).

(*S*)-7-Chloro-*N*-(1-oxo-1-(2-(4-(trifluoromethyl)phenyl)hydrazineyl)propan-2-yl)benzo[c][1,2,5]oxadiazole-4-sulfonamide, **17b**

Following the general procedure outlined, *tert*-butyl (*S*)-(1-oxo-1-(2-(4-(trifluoromethyl)phenyl)hydrazineyl)propan-2-yl)carbamate, **17a** (82 mg, 0.24 mmol) was transformed following flash chromatography into the title compound as a pale yellow solid (55 mg, 60%); R_f_ 0.34 (DCM/EtOH/NH_3_ [200:8:1]); m.p. 127–131 °C; υ_max_ (ATR) 3388 (N-H), 3315 (N-H), 3274, 2929, 2851, 1667 (C=O), 1619, 1323, 1159, 1110, 1067, 949, 836, 633, 597, 579, 511 cm^−1^; δ_H_ (700 MHz, DMSO-d_6_) 9.90 (1H, bs, N*H*), 8.80 (1H, d, *J* 8, N*H*), 8.24 (1H, bs, N*H*), 8.00 (1H, d, *J* 7, Ar-*H*), 7.82 (1H, d, *J* 7, Ar-*H*), 7.39 (2H, d, *J* 9, Ar-*H*), 6.61 (2H, d, *J* 9, Ar-*H*), 4.20 – 4.15 (1H, m, CH_3_C*H*CO), 1.26 (3H, d, *J* 7, C*H*_3_CHCO); δ_C_ (176 MHz, DMSO-d_6_) 171.4 (NHCH*C*O), 152.5 (Ar-*C*), 149.2 (Ar-*C*NO), 145.6 (Ar-*C*NO), 134.4 (Ar-*C*), 131.2 (Ar-*C*), 128.9 (Ar-*C*), 128.3 (Ar-*C*), 126.5 (Ar-*C*), 125.4 (Ar-*C*), 124.6 (*C*F_3_), 111.7 (Ar-*C*), 51.2 (NH*C*HCO), 19.8 (CH*C*H_3_); *m*/*z* (ES^+^) 464 ([^35^Cl] MH^+^), 466 ([^37^Cl] MH^+^); HRMS (ES^+^) Found [^35^Cl]MH^+^, 464.0400 (C_16_H_14_F_3_N_5_O_4_S^35^Cl requires 464.0402).

7-Chloro-N-(2-oxo-2-(2-phenylhydrazinyl)ethyl)benzo[c][1,2,5]oxadiazole-4-sulfonamide, **10**

Following the general procedure outlined, *tert*-butyl 2-oxo-2-(2-phenylhydrazinyl)ethylcarbamate, **18a** (64 mg, 0.24 mmol) was transformed following flash chromatography into the title compound as a yellow solid (41 mg, 55%); R_f_ 0.4 (DCM/MeOH 9:1) as a mixture of rotomers in the ratio [4:1] by NMR @ 25 °C; m.p. 209–211 °C; υ_max_ (ATR) 3262 (NH), 1738, 1647 (C=O), 1606, 1530, 1494, 1410, 1346, 1190, 1157, 947, 834, 747 cm^−1^; δ_H_ (500 MHz, DMSO-d_6_) 9.70 (1H, s, Ar-N*H*NH), 9.13 (0.24H, s, Ar-NHN*H*), 8.68 (1H, t, *J* 6, SO_2_N*H*), 8.35 (0.24H, t, *J* 6, SO_2_N*H*), 8.00 (1H, d, *J* 7, Ar-*H*), 7.94 (0.29H, d, *J* 7, Ar-*H*), 7.90 (0.27H, d, *J* 7, Ar-*H*), 7.84 (1H, d, *J* 7, Ar-*H*), 7.60 (1H, s, Ar-NHN*H*), 7.22 (0.55H, t, *J* 7, Ar-*H*), 7.10 (2H, t, *J* 7, Ar-*H*), 6.80 (0.25H, t, *J* 7, Ar-*H*), 6.71–6.66 (1.6H, m, Ar-*H*), 6.58 (2H, d, *J* 7, Ar-*H*), 3.96 (2H, d, *J* 6, NHC*H*_2_), 3.85 (2H, d, *J* 6, NHC*H*_2_); δ_C_ (126 MHz, DMSO-d_6_) 168.2 (*C*=O), 149.47 (Ar-*C*), 149.43 (Ar-*C*), 146.1 (Ar-*C*), 134.4 (Ar-*C*), 131.5 (Ar-*C*), 129.8 (Ar-*C*), 129.3 (Ar-*C*), 125.5 (Ar-*C*), 119.2 (Ar-*C*), 112.7 (Ar-*C*), 44.4 (NH*C*H_2_); *m*/*z* (ES^+^) 382 ([^35^Cl]MH^+^), 384 ([^37^Cl]MH^+^); HRMS (ES^+^) Found [^35^Cl]MH^+^, 382.03702 (C_14_H_13_^35^ClN_5_O_4_S requires 382.03713).

7-Chloro-*N*-(2-(2-(4-fluorophenyl)hydrazineyl)-2-oxoethyl)benzo[c][1,2,5]oxadiazole-4-sulfonamide, **19b**

Following the general procedure outlined, *tert*-butyl (2-(2-(4-fluorophenyl)hydrazineyl)-2-oxoethyl)carbamate, **19a** (68 mg, 0.24 mmol) was transformed following flash chromatography into the title compound as a brown solid (46 mg, 57%) as a mixture of rotamers [4:1] by NMR @ 25 °C; R_f_ 0.48 (DCM/EtOH/NH_3_ [200:8:1]); m.p. 180–185 °C; υ_max_ (ATR) 3364, 3245 (NH), 2918, 2850, 1673 (C=O), 1507, 1453, 1414, 1367, 1346 (S=O), 1216, 1157 (S=O), 1108, 1042 cm^−1^; NMR data given for major rotamer δ_H_ (700 MHz, DMSO-d_6_) 9.69 (1H, bs, Ar-N*H*), 8.63 (1H, t, *J* 9, CH_2_N*H*), 7.98 (1H, d, *J* 7, Ar-*H*), 7.81 (1H, d, *J* 7, Ar-*H*), 6.91 (2H, m, Ar-*H*), 6.56 (2H, m, Ar-*H*), 3.81 (2H, d, *J* 6, C*H*_2_NH); δ_F_ (376 MHz, DMSO) -126.45 (Ar-*F*); δ_C_ (176 MHz, DMSO-d_6_) 167.9 (*C*=O), 155.6 (Ar-*C*), 149.1 (Ar-*C*), 145.8 (Ar-*C*), 145.7 (Ar-*C*), 134.1 (Ar-*C*), 131.2 (Ar-*C*), 129.0 (Ar-*C*), 125.3 (Ar-*C*), 115.5 (Ar-*C*), 113.8 (Ar-*C*), 44.1 (*C*H_2_NH); *m*/*z* (ES^+^) 400 ([^35^Cl]MH^+^), 402 ([^37^Cl]MH^+^), 422 ([^35^Cl]MNa^+^), 424 ([^37^Cl]MNa^+^), 799 ([^35,35^Cl]2M + H^+^), 803 ([^35,37^Cl]2M + H^+^), 804 ([^37,37^Cl]2M + H^+^), 821 ([^35,35^Cl]2M + Na^+^), 823 ([^35,37^Cl]2M + Na^+^), 825 ([^37,37^Cl]2M + Na^+^); HRMS (ES^+^) Found [^35^Cl]MH^+^, 400.0271 (C_14_H_12_N_5_O_4_FS^35^Cl requires 400.0283).

7-Chloro-*N*-(2-(2-(4-chlorophenyl)hydrazineyl)-2-oxoethyl)benzo[c][1,2,5]oxadiazole-4-sulfonamide, **20b**

Following the general procedure outlined, *tert*-butyl (2-(2-(4-chlorophenyl)hydrazineyl)-2-oxoethyl)carbamate, **20a** (72 mg, 0.24 mmol) was transformed following flash chromatography into the title compound as a brown solid (35 mg, 43%) as a mixture of rotamers [6:1] by NMR @ 25 °C; R_f_ 0.37 (DCM/EtOH/NH_3_ [200:8:1]); m.p. 174–177 °C; υ_max_ (ATR) 3326, 3277 (NH), 3151, 3932, 2856, 1668 (C=O), 1523, 1491, 1325 (S=O), 1158 (S=O), 1107, 1067, 1043 cm^−1^; NMR data given for major rotamer δ_H_ (700 MHz, DMSO-d_6_) 9.72 (1H, bs, Ar-N*H*), 8.62 (1H, t, *J* 9, N*H*CH_2_), 7.98 (1H, d, *J* 7, Ar-*H*), 7.81 (1H, d, *J* 7, Ar-*H*), 7.72 (1H, bs, N*H*), 7.10 (2H, d, *J* 7, Ar-*H*), 6.55 (2H, d, *J* 7, Ar-*H*), 3.81 (2H, d, *J* 6, C*H*_2_NH); δ_C_ (176 MHz, DMSO-d_6_) 168.0 (*C*=O), 149.1 (Ar-*C*), 148.3 (Ar-*C*), 145.8 (Ar-*C*), 134.2 (Ar-*C*), 131.3 (Ar-*C*), 128.9 (Ar-*C*), 128.7 (Ar-*C*), 125.3 (Ar-*C*), 122.3 (Ar-*C*), 114.0 (Ar-*C*), 44.1 (*C*H_2_NH); m/z (ES^+^) 416 ([^35,35^Cl]MH^+^), 418 ([^35,37^Cl]MH^+^), 420 ([^37,37^Cl]MH^+^), 438 ([^35,35^Cl]MNa^+^), 440 ([^35,37^Cl]MNa^+^), 442 ([^37,37^Cl]MNa^+^), 853 ([^35,35,35,35^Cl]2M + Na^+^), 855 ([^35,35,35,37^Cl]2M + Na^+^), 857 ([^35,35,37,37^Cl]2M + Na^+^), 859 ([^35,37,37,37^Cl]2M + Na^+^), 861 ([^37,37,37,37^Cl]2M + Na^+^); HRMS (ES^+^) Found [^35,35^Cl]MH^+^, 415.998 (C_14_H_12_N_5_O_4_S^35^Cl_2_ requires 415.9987).

7-Chloro-*N*-(2-(2-(3-chlorophenyl)hydrazineyl)-2-oxoethyl)benzo[c][1,2,5]oxadiazole-4-sulfonamide, **21b**

Following the general procedure outlined, *tert*-butyl (2-(2-(3-chlorophenyl)hydrazineyl)-2-oxoethyl)carbamate, **21a** (72 mg, 0.24 mmol) was transformed following flash chromatography into the title compound as a pale yellow solid (29 mg, 22%) as a mixture of rotamers [5:1] by NMR @ 25 °C; R_f_ 0.29 (DCM/EtOH/NH_3_ [200:8:1]); m.p. 211–213 °C; υ_max_ (ATR) 3332, 3266, 3163 (NH), 2927, 2852, 1688 (C=O), 1597, 1524, 1469, 1425, 1371 (S=O), 1339, 1159 (S=O), 1073, 1043 cm^−1^; NMR data given for major rotamer δ_H_ (700 MHz, DMSO-d_6_) 9.75 (1H, bs, Ar-CN*H*), 8.66 (1H, t, *J* 6, CH_2_N*H*), 8.10 (1H, s, N*H*), 7.98 (1H, d, *J* 7, Ar-*H*), 7.82 (1H, d, *J* 7, Ar-*H*), 7.08 (1H, t, *J* 7, Ar-*H*), 6.68 (1H, ddd, *J* 8, 2, 1, Ar-*H*), 6.61 (1H, t, *J* 4, Ar-*H*), 6.52 (1H, ddd, *J* 8, 2, 1, Ar-*H*), 3.82 (2H, d, *J* 7, C*H*_2_NH); δ_C_ (176MHz, DMSO-d_6_) 168.13 (*C*=O), 150.9 (Ar-*C*), 149.2 (Ar-*C*), 145.8 (Ar-*C*), 134.1 (Ar-*C*), 131.2 (Ar-*C*), 130.7 (Ar-*C*), 128.9 (Ar-*C*), 125.3 (Ar-*C*), 118.4 (Ar-*C*), 114.0 (Ar-*C*), 111.8 (Ar-*C*), 111.1 (Ar-*C*), 44.2 (*C*H_2_NH); *m*/*z* (ES^+^) 416 ([^35,35^Cl]MH^+^), 418 ([^35,37^Cl]MH^+^), 420 ([^37,37^Cl]MH^+^), 471 ([^35,35^Cl]MNa + MeOH), 473 ([^35,37^Cl]MNa + MeOH), 475 ([^37,37^Cl]MNa + MeOH); HRMS (ES^+^) Found [^35,35^Cl]MH^+^, 415.9981 (C_14_H_12_N_5_O_4_S^35^Cl_2_ requires 415.9987).

7-Chloro-*N*-(2-(2-(2-chlorophenyl)hydrazineyl)-2-oxoethyl)benzo[c][1,2,5]oxadiazole-4-sulfonamide, **22b**

Following the general procedure outlined, *tert*-butyl (2-(2-(2-chlorophenyl)hydrazineyl)-2-oxoethyl)carbamate, **22a** (72 mg, 0.24 mmol) was transformed following flash chromatography into the title compound as a yellow solid (24 mg, 29%) as a mixture of rotamers [4:1] by NMR @ 25 °C; R_f_ 0.16 (DCM/EtOH/NH_3_ [600:8:1]); m.p. 173–175 °C; υ_max_ (ATR) 3324 (NH), 2928, 2851, 1669 (C=O), 1626, 1574, 1526, 1343, 1311, 1243, 1159 (S=O), 1088, 1042 cm^−1^; NMR data given for major rotamer δ_H_ (700 MHz, DMSO-d_6_) 9.85 (1H, bs, Ar-N*H*), 8.69 (1H, t, *J* 6, CH_2_N*H*), 8.00 (1H, d, *J* 7, Ar-*H*), 7.84 (1H, d, *J* 7, Ar-*H*), 7.27 (1H, s, C=ON*H*), 7.21 (1H, dd, *J* 8*,* 2, Ar-*H*), 7.10 (1H, dt, *J* 8, 7, Ar-*H*), 6.71 (1H, m, Ar-*H*), 6.58 (1H, dd, *J* 8*,* 1, Ar-*H*), 3.85 (2H, d, *J* 6, C*H*_2_NH); δ_C_ (176 MHz, DMSO-d_6_) 168.1 (*C*=O), 149.2 (Ar-*C*), 145.8 (Ar-*C*), 144.5 (Ar-*C*), 134.2 (Ar-*C*), 131.2 (Ar-*C*), 129.5 (Ar-*C*), 128.9 (Ar-*C*), 125.3 (Ar-*C*), 120.6 (Ar-*C*), 117.5 (Ar-*C*), 113.2 (Ar-*C*), 44.1 (*C*H_2_NH); m/z (ES^+^) 416 ([^35,35^Cl]MH^+^), 418 ([^35,37^Cl]MH^+^), 420 ([^37,37^Cl]MH^+^), 471 ([^35,35^Cl]MNa + MeOH), 473 ([^35,37^Cl]MNa + MeOH), 475 ([^37,37^Cl]MNa + MeOH); HRMS (ES^+^) Found [^35,35^Cl]MH^+^, 415.9972 (C_14_H_12_N_5_O_4_S^35^Cl_2_ requires 415.9987).

### 4.2. Biological Assessment

Bacterial strains and growth media used in this study (Appendix A).

#### 4.2.1. Bacterial Growth Inhibition Assays

The minimum inhibitory concentration of the compounds against all strains using stand REMA assay protocols [14]. Briefly, 100 μL of relevant growth media was added to all wells of a sterile 96-well plate (Corning Incorporated, Corning, NY, USA). The wells in rows A to H in columns 1 received 94.88 μL of growth medium (7H9 media was supplemented with 0.2% casamino acids, 24 μg/mL pantothenate and 10% OADC, Beckton Dickinson, Sparks, MD, USA). Compounds were added to rows A1-H1 (quadruplet per compound) followed by 1:2 serial dilutions across the plate to column 11 were 100 μL of excess medium was discarded from the wells in column 11. The bacterial cultures at 0.5 McFarland standard diluted 1:25 was added to the wells in rows A to H in columns 1 to 11 (100 μL), where the wells in column 12 served as drug-free controls (positive and negative). The plates were sealed with parafilm^TM^ and incubated at 37 °C, unless 30 °C was stated as the optimum for the organism. Freshly prepared filter sterilised resazurin (0.2% *w*/*v*, Sigma Aldrich, Dorset, UK) was filter sterilised and 10 μL added to all wells and re-incubated at 37 °C or 30 °C for 24 h or until the positive and negative controls showed a clear result.

#### 4.2.2. Mammalian Cytotoxicity Determination Using the MTT Assay

In vitro chemosensitivity of Human NCI-H460 lung carcinoma cells to the agents were determined using the MTT assay, described elsewhere [18]. Cells were exposed to the amino acid hydrazides or benzoxa-[2,1,3]-diazole amino acid hydrazides (10 μM), or solvent (dimethyl sulphoxide; DMSO) in quadruplicate. Solvent concentrations did not exceed 0.1% and were not cytotoxic. Chemosensitivity and cell survival was assessed following 96 h compound exposure, with cytotoxicity relative to vehicle control subsequently determined.

## 5. Conclusions

As shown by the present study, this interplay between cytotoxicity and antibacterial activity can be readily manipulated through the substitution patterns on each component, aromatic hydrazides, the size of the amino acid side chain and the benzoxa-[2,1,3]-diazole (Figure 3). The ease of manipulation makes this an attractive template and a full examination of all these parameters is the subject of ongoing efforts which will be reported in due course.

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
