# Peer review of "Identification of Novel Benzoxa-[2,1,3]-diazole Substituted Amino Acid Hydrazides as Potential Anti-Tubercular Agents"

_molecules, 2019, doi:10.3390/molecules24040811_

Round 1

Reviewer 1 Report

The manuscript describes the synthesis of new benzoxa-[2,1,3]-diazole substituted amino acid hydrazides and the evaluation of their anti-tubercular activity.

Although the data are interesting and the compounds fully characterized, the manuscript should be carefully rewritten, before acceptance, since it contains many mistakes such as:

Line 69: SI is missing, however the data seem to be presented in Table 1, page 3

Line 102: Figure S2 is missing

Lines 108 and 117 and Table 1 page 5: replace 13b-22b by 14b-22b. 13b=9

Line 110: Table S3 is missing

Line 119. Table 1 or 2?

Line 141: Figure 3 is not mentioned in the text

Line 284: 77? correct the name of the compound

Line 349: mp exceeds by far the acceptable range

Ref 1: website should be given

Author Response

The authors would like to thank the reviewer for their comments on our article. The reviewer has commented on the interest of the paper yet highlighted some minor corrections which we have addressed as follows:

Line 69: SI is missing, however, the data seem to be presented in Table 1, page 3

The full screen is highlighted in the supporting information, Table S1 whilst Table 1 in the manuscript is as a result of a dose range REMA assay.

Line 102: Figure S2 is missing

The VT experiment data is present in the supporting information document below the first table.

Lines 108 and 117 and Table 1 page 5: replace 13b-22b by 14b-22b. 13b=9

Please find the renumbering in the document at these positions and the supporting information Table S3

Line 110: Table S3 is missing

The full bacterial screen of the compounds is present in the supplementary information document

Line 119. Table 1 or 2?

Please find the correction in the manuscript

Line 141: Figure 3 is not mentioned in the text

Please find the correction in the manuscript

Line 284: 77? correct the name of the compound

Please find the correction in the manuscript

Line 349: mp exceeds by far the acceptable range

Please find a corrected m.p. in the manuscript

Ref 1: the website should be given

Please find the correction in the manuscript

Reviewer 2 Report

Comments are noted on the manuscript, a few minor edits are required. It would be useful to put the activity of these new compounds in context to existing drugs. Are they as active, or are they still an order of magnitude less active than current therapies. Is there anything known about the mode of activity? As mentioned in the introduction, drug resistance can occur fairly rapidly if the mode of activity is identical to current therapies. Elaborating on this could strengthen the argument why these compounds may be worth further scrutiny.

Author Response

Comments are noted on the manuscript, a few minor edits are required. It would be useful to put the activity of these new compounds in context to existing drugs. Are they as active, or are they still an order of magnitude less active than current therapies. Is there anything known about the mode of activity? As mentioned in the introduction, drug resistance can occur fairly rapidly if the mode of activity is identical to current therapies. Elaborating on this could strengthen the argument why these compounds may be worth further scrutiny.

The authors would like to thank the reviewer for their comments on our article. We hope that you will see from the changes made that we have worked to address all the minor edits required. Regarding the references and specifically the et al. notation, the bibliography was generated using the MDPI CSL file and as such adheres to the guidelines for this journal, see below:

·      For documents co-authored by a large number of persons (more than 10 authors), you can either cite all authors or cite the first ten authors, then add a semicolon and add ‘et al.’ at the end: 

Author 1; Author 2; Author 3; Author 4; Author 5; Author 6; Author 7; Author 8; Author 9; Author 10; et al. 

The reviewer has asked about the mode of activity of the compounds and to the best of our knowledge, there is no such available information. As mentioned in the paper, benzodiazoles have been designed to target HisG and APS reductase and so these may be the target of our compounds. Moreover, our previous studies with these molecules have shown that they target glutathione and as such mycothiolates might be our target. In any case, the authors feel that the addition of this commentary would dilute the impact of the paper and complicate the discussion. In addition, identify the target(s) of these molecules through genomics, proteomics and metabolomic studies is the subject of our continued study and will be reported in due course. 

Reviewer 3 Report

The manuscript attempts to report on anti-tubercular activity of novel benzoxadiazole-derived amino acid hydrazides. Although overall interesting idea is presented in an introduction section, the manuscript suffers from a large number of shortcomings, which hinder the possibility of its acceptance for publication.

1.       The abstract requires a substantial rewriting. In its current form it sounds more like a paragraph from an introduction chapter rather than the proper abstract. Short and informative description of metodologies applied and the results obtained, (the synthetic pathway, biological assays used and the activities of compounds tested) should be implemented in the abstract.

2.       The antibacterial activity results, and especially Tables 1, 2,and 3 suffer from a large number of inconsistencies. The title of manuscript suggests the extended investigation into anti-tubercular activity of compounds obtained, while the data in Tables refer to activities against various species.

3.       Table 2 refers to „Results of the synthesis” while the biological activities are listed therein. No chemical nor physical data except yields and rotamer ratios are presented in this table.

4.       The structures of novel compounds do not vary much. The structural variety relies mainly on the aminoacid used. The hydrazine moiety was in most cases p-CF3 – substituted. Therefore the discussion of structure-activity relationship of these compounds related to their anti-tubercular activity is insufficient.

5.       Table 3 does not provide any information regarding the anti-tubercular activity of compounds tested and hence should be moved to supplementary data.

6.       Since the antibacterial activity results expressed as MIC are highly questionable the anti-tubercular activity should be expressed as MIC50 or MIC90. The discussion of structure-activity relationship is therefore unreliable and requires much more accurate elaboration. The biological results presented are insufficient.

7.       Large number of reference citations is missing, variuos font sizes and types are used, italics are not applied properly. This creates an impression of carelessness and is unacceptable in high impact journals such as Molecules.

8.       Materials and methods section is missing in the manuscript. This section must be implemented, especially when authors report on the synthesis of novel compounds.

Given the large number of shortcomings the manuscript in its current form should not be accepted for publication and hence my recommendation is to reject this manuscript. 

Author Response

The manuscript attempts to report on anti-tubercular activity of novel benzoxadiazole-derived amino acid hydrazides. Although overall interesting idea is presented in an introduction section, the manuscript suffers from a large number of shortcomings, which hinder the possibility of its acceptance for publication.

The authors would like to thank the reviewer for their comments on our article. We hope that you will see from the changes made that we have worked to address all the modification requested.

1.       The abstract requires a substantial rewriting. In its current form it sounds more like a paragraph from an introduction chapter rather than the proper abstract. Short and informative description of metodologies applied and the results obtained, (the synthetic pathway, biological assays used and the activities of compounds tested) should be implemented in the abstract.

The reviewer should find that editing of the abstract has been undertaken to shorten and include the information as outlined above. Hopefully, it now reflects the content of the paper more accurately.

2.       The antibacterial activity results, and especially Tables 1, 2,and 3 suffer from a large number of inconsistencies. The title of manuscript suggests the extended investigation into anti-tubercular activity of compounds obtained, while the data in Tables refer to activities against various species.

The reviewer has highlighted that whilst the title suggests a primary focus on the anti-tubercular activity the authors undertook an antimicrobial screen to demonstrate species specificity. Partial results are included in Table 1 for direct MIC determination from the initial fixed concentration screen included in Table S1 of the Supplementary data. The remaining data displayed in the manuscript is focused on Mtbwhilst Supplemental Table S3 reinforces species specificity as another complete antibacterial screen was performed on the second iteration of compounds.

3.       Table 2 refers to „Results of the synthesis” while the biological activities are listed therein. No chemical nor physical data except yields and rotamer ratios are presented in this table.

Having reviewed the caption for table 2 the authors believe that the text included is appropriate for the data displayed. The authors are unclear as to what additional chemical or physical data should be presented in the table and would seek further discussion on this if required. The authors feel that the duplication of chemical analyses from the materials and methods is unnecessary and would reduce the impact of the data presented in the table. 

4.       The structures of novel compounds do not vary much. The structural variety relies mainly on the aminoacid used. The hydrazine moiety was in most cases p-CF– substituted. Therefore the discussion of structure-activity relationship of these compounds related to their anti-tubercular activity is insufficient.

The authors believe that larger variations in structural characteristics would suffer from too much speculation around a SAR analysis without having the greater detail gained from a larger pool of molecules. This study presents a small, tightly grouped set of molecules to allow for the best SAR analyse to be performed. The studies primary focus is on the minor modifications to understand the biological activity of the functional groups in relation to the overall compound.  

5.       Table 3 does not provide any information regarding the anti-tubercular activity of compounds tested and hence should be moved to supplementary data.

 Table 3 has been moved to the SI information for clarity of the paper

6.       Since the antibacterial activity results expressed as MIC are highly questionable the anti-tubercular activity should be expressed as MIC50 or MIC90. The discussion of structure-activity relationship is therefore unreliable and requires much more accurate elaboration. The biological results presented are insufficient.

The reviewer comments that the antibacterial activity results are highly questionable, we respectfully disagree. The results have been generated consistently following an end-point determination REMA assay in quadruplet, a well-established method which has been published on multiple occasions by these authors and others. 

7.       Large number of reference citations is missing, variuos font sizes and types are used, italics are not applied properly. This creates an impression of carelessness and is unacceptable in high impact journals such as Molecules.

The submitted manuscript has observed minor editorial issues upon transfer to the journal template which has now been corrected. 

8.       Materials and methods section is missing in the manuscript. This section must be implemented, especially when authors report on the synthesis of novel compounds.

Whilst the detail required to repeat the work is presented in the Experimental section, the authors have now renamed this as the Materials and Methods section to align with the journal layout.

Given the large number of shortcomings the manuscript in its current form should not be accepted for publication and hence my recommendation is to reject this manuscript. 

Reviewer 4 Report

The present study by Brown et al. describes the design and synthesis of benzoxa-[2,1,3]-diazole substituted amino acid hydrazides as potential anti-tubercular agents. However, some revision should be done before publication.

1. How about the purities of test compounds? This raises a question mark about their purity and hence relevance and reliability of the biological results.

2. A supplementary material for publication online should be provided by the authors. It should contain representative 1H and 13C NMR spectra.

Author Response

The present study by Brown et al. describes the design and synthesis of benzoxa-[2,1,3]-diazole substituted amino acid hydrazides as potential anti-tubercular agents. However, some revision should be done before publication.

1. How about the purities of test compounds? This raises a question mark about their purity and hence relevance and reliability of the biological results.

2. A supplementary material for publication online should be provided by the authors. It should contain representative 1H and 13C NMR spectra.

The authors would like to thank the reviewer for their comments on our article. We hope that you will see from the changes made that we have worked to address all the minor edits required. Included in the SI information are all the 1H NMR and 13C NMR for the compounds presented in this report. Moreover, where possible we have included an HPLC trace for new compounds to highlight the purity in the hope that this alleviates the concerns raised.

Round 2

Reviewer 3 Report

The overall quality of the manuscript is now substantially improved. The explanations provided by the authors in their rebuttal letter are satisfactory. Therefore I am happy to recommend the manuscript for publication.